# OPEN-VOCABULARY PANOPTIC/UNIVERSAL SEGMENTATION WITH MASKCLIP

## ABSTRACT

In this paper, we tackle an emerging computer vision task, open-vocabulary panoptic segmentation, that aims to perform panoptic segmentation (background semantic labeling + foreground instance segmentation) for arbitrary categories of text-based descriptions in inference time. We first build a baseline method by directly adopting pre-trained CLIP models without finetuning nor distillation. We then develop MaskCLIP, a Transformer-based approach with a MaskCLIP Visual Encoder, which is an encoder-only module that seamless integrates mask tokens with a pre-trained ViT CLIP model for semantic/instance segmentation and class prediction. MaskCLIP learns to efficiently and effectively utilize pre-trained dense/local CLIP features within the MaskCLIP Visual Encoder that avoids the time-consuming student-teacher training process. We obtain encouraging results for open-vocabulary panoptic/instance segmentation and state-of-the-art results for semantic segmentation on ADE20K and PASCAL datasets. We show qualitative illustration for MaskCLIP with online custom categories.

## 1 INTRODUCTION

Panoptic segmentation (Kirillov et al., 2019b) or image parsing (Tu et al., 2005) integrates the task of semantic segmentation (Tu, 2008) for background regions (e.g. "stuff" like "road", "sky") and instance segmentation (He et al., 2017) for foreground objects (e.g. "things" such as "person", "table"). Existing panoptic segmentation methods (Kirillov et al., 2019b;a; Li et al., 2019; Xiong et al., 2019; Lazarow et al., 2020) and instance segmentation approaches (He et al., 2017) deal with a fixed set of category definitions, which are essentially represented by categorical labels without semantic relations. DEtection TRansformer (DETR) (Carion et al., 2020) is a pioneering work that builds a Transformer-based architecture for both object detection and panoptic segmentation.

The deep learning field is moving rapidly towards the open-world/zero-shot settings (Bendale & Boult, 2015) where computer vision tasks such as classification (Radford et al., 2021), object detection (Li et al., 2022b; Zareian et al., 2021; Zang et al., 2022; Gu et al., 2022; Cai et al., 2022), semantic labeling (Li et al., 2022a; Ghiasi et al., 2022), and image retrieval (Bendale & Boult, 2015; Hinami & Satoh, 2018; Zareian et al., 2021; Hinami & Satoh, 2018; Kamath et al., 2021) perform recognition and detection for categories beyond those in the training set.

In this paper, we take the advantage of the existence of pre-trained CLIP image and text embedding models (Radford et al., 2021), that are mapped to the same space. We first build a baseline method for open-vocabulary panoptic segmentation using CLIP models without training. We then develop a new algorithm, MaskCLIP, that is a Transformer-based approach efficiently and effectively utilizing pre-trained dense/local CLIP features without heavy re-training. The key component of MaskCLIP is a Relative Mask Attention (RMA) module that seamlessly integrates the mask tokens with a pre-trained ViT-based CLIP backbone. MaskCLIP is distinct and advantageous compared with existing approaches in three aspects: 1) A canonical background and instance segmentation representation by the mask token representation with a unique encoder-only strategy that tightly couples a pre-trained CLIP image feature encoder with the mask token encoder. 2) MaskCLIP avoids the challenging student-teacher distillation processes such as OVR-CNN (Zareian et al., 2021) and ViLD (Gu et al., 2022) that face limited number of teacher objects to train; 3) MaskCLIP also learns to refine masks beyond simple pooling in e.g. OpenSeg (Ghiasi et al., 2022).

The contributions of our work are listed as follows.

- We develop a new algorithm, MaskCLIP, to perform open-vocabulary panoptic segmentation building on top of canonical background and instance mask representation with a cascade mask proposal and refinement process.
- We device the MaskCLIP Visual Encoder under an encoder-only strategy by tightly coupling a pre-trained CLIP image feature encoder with the mask token encoder, to allow for the direct formulation of the mask feature representation for semantic/instance segmentation+refinement, and class prediction. Within the MaskCLIP Visual Encoder, there is a new module called Relative Mask Attention (RMA) that performs mask refinement.
- MaskCLIP expands the scope of the existing CLIP models to open-vocabulary panoptic segmentation by demonstrating encouraging and competitive results for open-vocabulary- panoptic, instance, and semantic segmentation.

## 2 RELATED WORK

Table 1: Comparison for recent open-vocabulary approaches for object detection, semantic segmentation, instance segmentation, and panoptic segmentation. GLIP (Li et al., 2022b); OVR-CNN (Zareian et al., 2021); ViLD (Gu et al., 2022); RegionCLIP (Zhong et al., 2022); OV-DETR (Zang et al., 2022); LSeg (Li et al., 2022a); OPenSeg (Ghiasi et al., 2022); DenseCLIP (Rao et al., 2022); XPM (Huynh et al., 2022). ✘ indicates that the corresponding method is loosely following the definition. Dense Clip features refer to the use of pixel-wise/local features. Note that OpenSeg uses its ALIGN (Jia et al., 2021), which is an alternative to CLIP.

| Task | Method | Arbitrary Online Inference | Segmentation semantic | instance | Dense CLIP features | Training data | Annotation type |
|---|---|---|---|---|---|---|---|
| Object Det. | GLIP | ✔ | | | | FourODs, GoldG, Cap24M | labels + bbox + captions |
| | OVR-CNN | ✔ | | | | COCO base, CC3M | bbox + captions |
| | ViLD | ✔ | | | | COCO | labels + bbox |
| | RegionCLIP | ✔ | | | | CC3M, COCO | captions |
| Semantic Seg. | LSeg | ✘ | ✔ | | | COCO + others | labels + segmentations |
| | OpenSeg | ✔ | ✔ | ✘ | ✔ | COCO, LocalizedNarratives | masks + captions |
| | DenseCLIP | | ✔ | | ✔ | COCO | labels + segmentations |
| Instance Seg. | XPM | ✘ | | ✔ | | COCO, CC3M | labels + masks + captions |
| Panoptic Seg. | MaskCLIP (ours) | ✔ | ✔ | ✔ | ✔ | COCO | labels + masks |

**Open vocabulary.** The open vocabulary setting is gaining increasing popularity lately as traditional fully supervised setting cannot handle unseen classes during testing, while real world vision applications like scene understanding, self driving and robotics are commonly required to predict unseen classes. Previous open-vocabulary attempts have been primarily made for object detection. ViLD (Gu et al., 2022) trains a student model to distill the knowledge of CLIP. RegionCLIP (Zhong et al., 2022) finetunes the pretrained CLIP model to match the image areas with corresponding texts. OV-DETR (Zang et al., 2022) uses CLIP as an external model to obtain the query embedding from CLIP model. Recently there is also work made for open-vocabulary semantic segmentation (Ghiasi et al., 2022).

**Panoptic segmentation.** Existing panoptic segmentation (or image parsing (Tu et al., 2005)) methods (Kirillov et al., 2019b;a; Li et al., 2019; Xiong et al., 2019; Lazarow et al., 2020) perform training and testing based on a fixed set of category labels. Open-set panoptic segmentation (Hwang et al., 2021) is an exemplar based approach that requires categories to be known in advance, which is narrower than the open-vocabulary setting where categories of interest can be freely specified in inference.

**Open-vocabulary panoptic segmentation: an emerging task.** As open-set, open-world, zero-shot, and open-vocabulary are relatively new concepts that have no commonly accepted definitions, thus, different algorithms are often not directly comparable with differences in problem definition/setting, training data, and testing scope. Table 1 gives a summary for the recent open-vocabulary applications. XPM (Huynh et al., 2022) utilizes vision-language cross modal data to generate pseudo-mask supervision to train a student model for instance segmentation, and thus, it may not be fully open-vocabulary to allow for arbitrary object specifications in the inference time. LSeg (Li et al., 2022a) also has limited open-vocabulary aspect as the learned CNN image features in LSeg are not exposed to representations beyond the training labeling categories. OpenSeg (Ghiasi et al., 2022) is potentially applicable for instance/panoptic segmentation, but OpenSeg is formulated to be trained on captions which lack instance-level information that is fundamental for panoptic segmentation. The direct image feature pooling strategy in OpenSeg is potentially another limiting factor towards the open-vocabulary panoptic segmentation. Nevertheless, no results for open-vocabulary panoptic/instance segmentation are reported in (Ghiasi et al., 2022).

**CLIP model distillation/reuse.** After its initial release, the CLIP model (Radford et al., 2021) that is learned from large-scale image-text paired captioning datasets has received a tremendous amount of attention. Some other similar vision-language models have also been proposed later e.g. ALIGN (Jia et al., 2021), GLIP (Li et al., 2022b). Many algorithms have been developed lately (Zang et al., 2022; Wang et al., 2022; Zhong et al., 2022; Luo et al., 2021; Patashnik et al., 2021; Shen et al., 2022) trying knowledge distillation from the CLIP model to benefit the down-stream tasks one way or the other by leveraging the rich semantic language information paired in the images. Here, we directly adopt the backbone of CLIP image model to train for open-vocabulary panoptic segmentation. There have been attempts (Rao et al., 2022; Zhou et al., 2022) that use the dense CLIP features to represent pixel-wise feature as teacher model to train student model for semantic segmentation.

## 3 METHOD

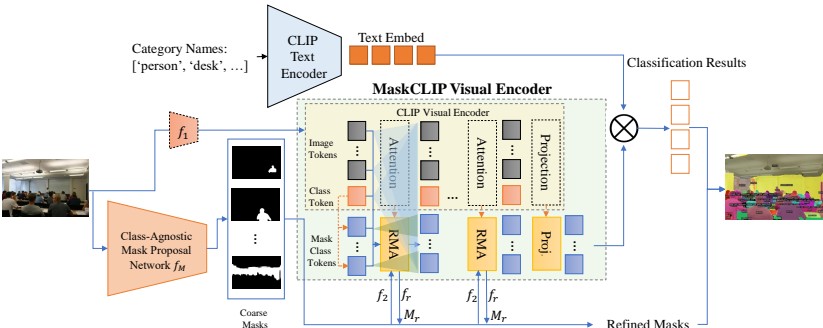

Figure 1: **Illustration of the pipeline.** Our pipeline contains two stages. The first stage is a class-agnostic mask proposal network and the second stage is built on the pretrained CLIP ViT model. All the weights of the CLIP ViT model during training are fixed. Arrows in orange denote weight sharing. The embeddings' weights of Mask Class Tokens are shared by Class Tokens in the CLIP ViT model and are fixed. RMA represents Relative Mask Attention which is built based on the CLIP ViT attention layer. RMA contains all the weights from CLIP ViT attention layer which are all fixed during training. Additional weights are added in RMA for further mask information utilization and mask refinement. The demo image we use here is from ADE20K (Zhou et al., 2019).

Our pipeline, shown in Figure 1, contains two stages. The first stage is a class-agnostic mask proposal network. The second stage is MaskCLIP Visual Encoder which is built on the CLIP (Radford et al., 2021) ViT architecture. It takes the images and the coarse masks from the first stage as the input and will output refined masks along with the corresponding dense image features for further classification using the text embeddings from the CLIP text encoder.

### 3.1 CLASS-AGNOSTIC MASK PROPOSAL NETWORK

Our Class-Agnostic Mask Proposal Network is built on instance/segmentation models such as MaskRCNN(He et al., 2017) and Mask2Former(Cheng et al., 2022). To make the model class-agnostic, we remove the class supervision during training. The classification head thus becomes a binary classification for either positive or negative in these models.

### 3.2 MASKCLIP VISUAL ENCODER

Similar to CLIP, our MaskCLIP Visual Encoder also predicts the image features. Unlike the CLIP Visual Encoder that only uses on class token to output the feature of the whole image. Our MaskCLIP Visual Encoder uses another $M$ Mask Class Tokens to output the partial/dense features for each corresponding area of the image given the masks. The Mask Class Tokens use attention masks and Relative Mask Attention to obtain the partial/dense features which we discuss in the following two parts.

#### 3.2.1 MASK CLASS TOKENS

In order to obtain dense image features for the corresponding masks or bounding boxes for further recognition or distillation, an easy way to do this is simply masking or cropping the image and then sending the obtained image to the pretrained image encoder. This method has been widely used in

several open vocabulary object detection methods (Zhong et al., 2022; Gu et al., 2022). The problem is that it's not computation efficient ($N$ masks/boxes will lead to $N$ images and they will be computed through the image encoder independently) and also loses the ability to see the global image context information which is very important for recognizing some objects and stuff. For masking, another problem is that masks are in different shapes and simply masking the image will cause the resulted image to have transparent background which usually doesn't exist in real images that are used for training in large language-vision models e.g. CLIP.

To solve this, we propose Mask Class Tokens for efficient feature extraction from images without losing the global image context information. In the original CLIP ViT-based visual encoder framework, the input of the network is $N$ image tokens and $1$ class token. The final output of the class token will be used for the relation computation with the text embeddings. Our newly introduced $M$ Mask Class Tokens will be alongside with the image tokens and the class token. The embeddings' weights of the Mask Class Token are provided by the class token in the pretrained CLIP ViT model and are fixed. Each Mask Class Token will output a corresponding dense image feature similar to the class token which outputs the feature of the whole image. To achieve this, we design an attention mask as following

$$\mathcal{M} = \begin{bmatrix} \mathcal{F}_{(N+1)\times(N+1)} & \mathcal{T}_{(N+1)\times M} \\ \mathcal{M}'_{M\times N} \quad \mathcal{F}_{M\times 1} & \mathcal{T}_{M\times M} \end{bmatrix} \tag{1}$$

in which $M$ is the number of Mask Class Tokens, $N$ is the number of image tokens, $\mathcal{T}_{m\times n}$ is an $m \times n$ True matrix, $\mathcal{F}_{m\times n}$ is an $m \times n$ False matrix and $\mathcal{M}'$ is defined as following:

$$\mathcal{M}'_{i,j} = \begin{cases} \text{False} & \text{if } mask_i \text{ contains at least one pixel in } patch_j \\ \text{True} & \text{otherwise.} \end{cases} \tag{2}$$

where True means that this position is masked out i.e. not allowed to attend and False otherwise.

In our mask attention matrix $\mathcal{M}$, $\mathcal{F}_{(N+1)\times(N+1)}$ shows the $N$ Image Tokens and one Class Token are attending each other as in the original CLIP. $\mathcal{T}_{(N+1)\times M}$ shows that the $N$ Image Tokens and one Class Token are not attending the $M$ Mask Class Tokens. $\mathcal{M}'_{M\times N}$ shows that the Mask Class Tokens are attending the Image Tokens given the corresponding masks. $\mathcal{F}_{M\times 1}$ shows that the $M$ Mask Class Tokens are attending the Class Token. $\mathcal{T}_{M\times M}$ shows that the $M$ Mask Class Tokens are not interacting with each other.

In this way, each Mask Class Token will learn from the corresponding mask area of the images. The image tokens are also interacting with each other which means the global information won't lose. And it's also very efficient since we don't need to do redundant computing for each mask or finetune the pretrained model. However, the mask information are not fully utilized and they cannot be refined either. But we will see in the experiments later that simply adpoting Mask Class Tokens to the pretrained CLIP model without any finetuning will already serve as a competitive baseline.

### 3.2.2 RELATIVE MASK ATTENTION

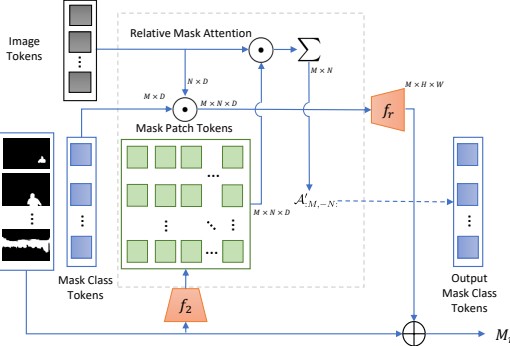

Figure 2: **Relative Mask Attention.** Our Relative Mask Attention mechanism adds another attention matrix $A'_{:M,-N:}$ to the original attention matrix. The newly added attention matrix is computed using the Image Tokens and the Mask Patch Tokens. The mask patch tokens are acquired by patchifying the masks using the similar way for the images as showed here. Moreover, the masks are refined by using $M_r$ in Eq. 5 which is computed by Image Tokens and Mask Class Tokens.

To further utilize the mask information and refine the coarse masks, we propose Relative Mask Attention mechanism in our transformer. Our key design principle is try not to change the CLIP features directly as this would destroy the learned relationship between the image features and text features in the CLIP model. Therefore, we adopt a way to only change the attention matrix in the transformer to learn a better linear combination of the values in the attention layers according to the mask information. As in Figure 2, our proposed Relative Mask Attention Mechanism only changes the attention matrix and refines the masks. $M_r$ is defined in Eq. 5. $\mathcal{A}'_{:M,-N:}$ is defined in Eq. 3. $f_M$ is the class-agnostic mask proposal network. $f_1$ and $f_2$ are two downsampling networks that encode the images/masks to image tokens/mask patch tokens sharing the same architecture. $f_r$ is a two-layer convolutional network that maps the attention matrix to a mask residual.

Similar to relative positional encoding, we use a relative attention mechanism here. Let $D$ be the dimension of the token embedding, for each Mask Class Token $T_i^{\mathrm{MC}} \in \mathbb{R}^D$ with a corresponding mask $K_i \in \mathbb{R}^{H \times W}$ whose shape is the same as the image, we use a similar way as for the images to get mask patch tokens $T^{\mathrm{MP}} \in \mathbb{R}^{M \times N \times D}$ in the computation of the attention. In our attention matrix, the Mask Class Tokens attending image tokens part will then be as following:

$$\mathcal{A}'_{:M,-N:} = \sum_c^D (\phi_{Q_m}(T^{\mathrm{MP}}) \odot \phi_{K_m}(T^{\mathrm{IM}}))_c \tag{3}$$

$$\mathcal{A}_{:M,-N:} = \frac{\phi_Q(T^{\mathrm{MC}}) \cdot \phi_K(T^{\mathrm{IM}}) + \mathcal{A}'_{:M,-N:}}{2\sqrt{D}} \tag{4}$$

where $T^{\mathrm{IM}} \in \mathbb{R}^{N \times D}$ is image tokens, $T^{\mathrm{MC}} \in \mathbb{R}^{M \times D}$ is Mask Class Tokens, $T^{\mathrm{MP}} \in \mathbb{R}^{M \times N \times D}$ is Mask Patch Tokens $\phi_Q, \phi_K, \phi_{Q_m}, \phi_{K_m}$ are linear transformations, $\odot$ is element-wise product and $\sum_c^D (\cdot)_c$ is the sum of the embedding dimension. $\phi_{K_m}(T^{\mathrm{IM}}) \in \mathbb{R}^{N \times D}$ will first be broadcast to $\mathbb{R}^{M \times N \times D}$ before doing element-wise production.

The attention will also in turn be used for the refinement of the masks. The vanilla attention can be seen as a relationship between each mask area and all the image patches. Thus we utilize this to help our coarse masks be more accurate. The updating process of the masks is as following:

$$M_r = \sigma(\sigma^{-1}(M_c) + f_r(\phi_Q(T^{\mathrm{MC}}) \odot \phi_K(T^{\mathrm{IM}}))) \tag{5}$$

where $M_c, M_r \in \mathbb{R}^{N \times H \times W}$ denotes the coarse mask and refined mask respectively, $f_r$ is a learnable non-linear function that maps the attention matrix to a mask residual, $\sigma$ and $\sigma^{-1}$ are sigmoid and inverse sigmoid functions respectively.

## 4 EXPERIMENTS

In this part, we train our proposed MaskCLIP method using COCO (Lin et al., 2014) training data and test on other datasets (ADE20K (Zhou et al., 2019; 2017), PASCAL Context (Mottaghi et al., 2014), LVIS) under the open vocabulary setting. Due to the novel setting of the open vocabulary panoptic segmentation task, we also compare our performance on open vocabulary semantic segmentation with previous methods. Apart from the quantitative results, we also provide qualitative results to validate our method that has good ability to learn dense image features and can support user-specified arbitrary categories.

### 4.1 DATASETS

**COCO:** COCO (Lin et al., 2014) includes 133 classes where 80 classes are things and 53 classes are stuff or background. There are 118k training images and 5k validation images. In our experiments, we first train the class-agnostic mask proposal network on COCO training dataset using the annotations of panoptic masks. Then we train our models on COCO training images in a supervised manner.

**ADE20K:** ADE20K (Zhou et al., 2019; 2017) contains 20,210 images and annotations for training and 2000 images and annotations for validation. It serves both panoptic segmentation and semantic segmentation. The full version (A-847) (Zhou et al., 2019) includes 847 classes and the short version (A-150) (Zhou et al., 2017) includes 150 classes. We use the validation set in ADE20K for

testing without any training on this dataset in which case we can test our model's capability of open vocabulary segmentation.

**PASCAL Context:** PASCAL Context (Mottaghi et al., 2014) contains 10,103 per-pixel annotations for images of PASCAL VOC 2010 (Everingham et al.), where 4998 for training and 5105 for validation. The full version (P-459) includes 459 classes and the short version includes 59 classes. This dataset serves as another benchmark testing our model's open vocabulary segmentation abiltiy.

**LVIS:** LVIS (Gupta et al., 2019) contains 100,170 images for training and 19,809 images for validation. It extends COCO (Lin et al., 2014) but contains 1,203 categories. It is considered as one of the most challenging benchmark for instance segmentation because of its large vocabulary, long-tailed distribution, and fine-grained classification. We report our model's performance of open vocabulary instance segmentation on the validation dataset.

## 4.2 IMPLEMENTATION DETAILS

**Class-Agnostic Mask Proposal Network.** In our first stage, we train a class-agnostic mask proposal network using MaskRCNN (He et al., 2017) and Mask2Former (Cheng et al., 2022) on COCO training data. The experiment setting we use for MaskRCNN is R50-FPN-1x. The backbone we use in Mask2Former is ResNet-50. All the training setting follows the default in their models.

**CLIP Baseline.** We design our first baseline by directly using the class-agnostic mask proposal network from the first stage and the pretrained CLIP model. We mask the images according to the masks from the class-agnostic mask proposal network and send the masked images to the CLIP model to get classification results. The pretrained CLIP model we use is ViT-L/14@336px and the text inputs we use are simply the category names defined by each dataset. Those two settings keep the same with the following two methods for fair comparison.

**MaskCLIP w/o RMA Baseline.** Our second baseline is based on the Mask Class Tokens which doesn't use the Relative Mask Attention mechanism. Instead of masking the images and sending the resulted images directly to the CLIP model for feature extraction, we use Mask Class Tokens to acquire the corresponding dense image features. The obtained image features will then be used for further open vocabulary classification.

The two baselines above don't need any training in the second stage and can be used to directly perform the open vocabulary tasks. We will demonstrate that the second baseline is better at feature extraction in both quantitative results and qualitative results under the open vocabulary setting and show the effectiveness and efficiency of the proposed Mask Class Tokens.

**MaskCLIP.** In our MaskCLIP method, we still use the CLIP ViT-L/14@336px pretrained model as with the previous two. This model has 24 attention layers and we add Relative Mask Attention in four of them which is 6, 12, 18, 24. We use AdamW (Loshchilov & Hutter, 2019) as our optimizer and the learning rate is set to 0.0001. We train our model on COCO training data for 10k iterations with a batchsize of 8. The training takes around 3h on 8 Nvidia A5000 GPUs.

**Loss Function.** The loss function is $\mathcal{L} = \lambda_{ce}\mathcal{L}_{ce} + \lambda_{dice}\mathcal{L}_{dice} + \lambda_{bce}\mathcal{L}_{bce}$, where $\mathcal{L}_{ce}$ is the loss for classification, $\mathcal{L}_{dice}$ and $\mathcal{L}_{bce}$ are the losses for mask localization. In our experiments, We set $\lambda_{ce} = 2, \lambda_{dice} = 5, \lambda_{bce} = 5$.

In next three parts, we evaluate our methods on open vocabulary panoptic, instance segmentation, and semantic segmentation tasks. The class-agnostic mask proposal networks we use in those methods are trained using Mask2Former other than noted.

## 4.3 OPEN-VOCABULARY PANOPTIC SEGMENTATION

First, we compare our MaskCLIP with the two baselines on ADE20K validation set under the open vocabulary panoptic segmentation setting. The results are presented in Table 2. As can be seen from the table, the MaskCLIP w/o RMA baseline performs better on all the metrics in panoptic segmentation setting which demonstrates that our feature extraction method is better than the vanilla way in this setting. It extracts the features without the need to changing the input and can simultaneously extract multiple mask area features easily. For 100 masks' feature extraction in a single image, the CLIP baseline takes ~3s on a single 3090 GPU while the MaskCLIP w/o RMA

baseline only takes ~0.6s which is **~4x** faster. Our MaskCLIP beats both baselines significantly as it utilizes the accurate mask information and refines the masks during the feature extraction process.

Table 2: **Results on open-vocabulary panoptic segmentation using the ADE20k validation dataset.** th and st represent thing and stuff classes respectively.

| Method | PQ ↑ | PQ$^{th}$ ↑ | PQ$^{st}$ ↑ | SQ ↑ | SQ$^{th}$ ↑ | SQ$^{st}$ ↑ | RQ ↑ | RQ$^{th}$ ↑ | RQ$^{st}$ ↑ |
|---|---|---|---|---|---|---|---|---|---|
| CLIP Baseline | 8.207 | 8.473 | 7.675 | 53.124 | 52.661 | 54.048 | 10.534 | 10.883 | 9.835 |
| MaskCLIP w/o RMA | 9.565 | 8.922 | 10.852 | 62.507 | 62.268 | 62.985 | 12.645 | 11.758 | 14.418 |
| MaskCLIP (MaskRCNN) | 12.860 | 11.242 | 16.095 | 64.008 | 64.183 | 63.658 | 16.803 | 14.968 | 20.473 |
| MaskCLIP | **15.121** | **13.536** | **18.290** | **70.479** | **70.021** | **71.396** | **19.211** | **17.448** | **22.737** |

In this part, we show two sets of images to demonstrate our model capability. The first is the qualitative results on ADE20K. We compare our method with the two baselines in Figure 3. It can be seen that our method performs much better than the two baselines. The results from the first column show that due to the lack of global information, CLIP baseline fails to predict the floor. Instead it predicts skyscraper. This is an easy case but if only floor area is provided it does have some similarities with the wall of a skyscraper. While the MaskCLIP w/o RMA baseline and MaskCLIP model can predict the floor correctly with the the global image context information.

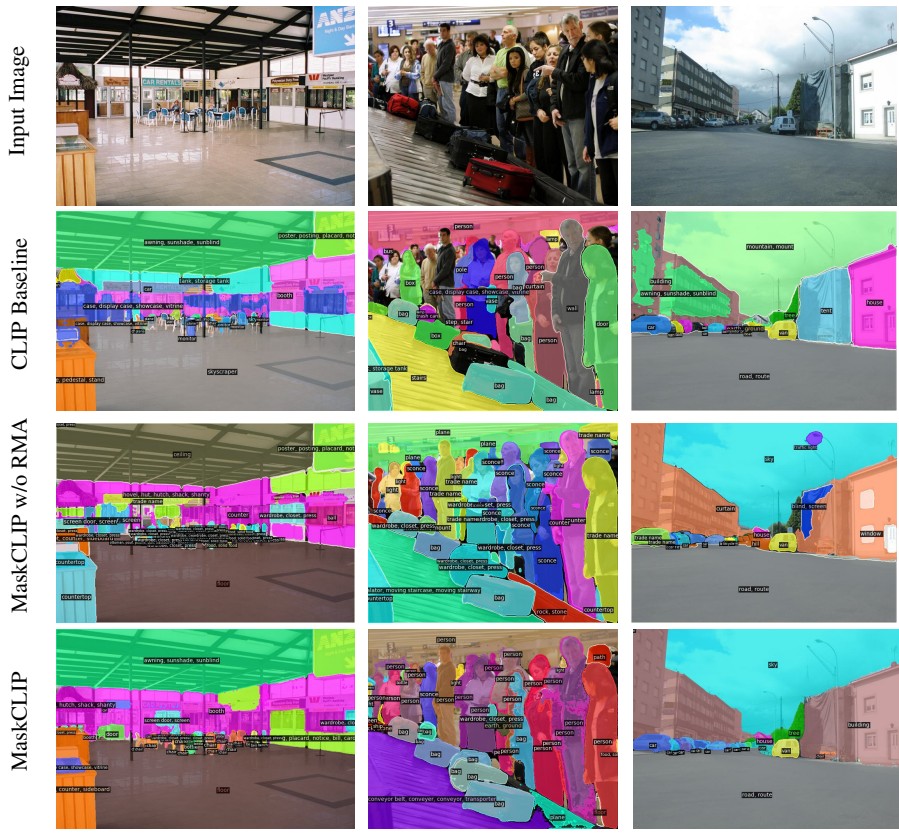

Figure 3: **Qualitative results on ADE20K panoptic segmentation.** The images are taken from the ADE20K validation set. We use the class names directly from the ADE20K 150 classes as the text inpputs. Three images are presented here using our MaskCLIP model along with the two baselines.

The second set of images we're presenting are in Figure 4. These figures show our capability of specifying any arbitrary classes in performing panoptic segmentation task. The results show that though we train a new model based on the CLIP model without any distillation methods, we can still preserve the CLIP image features very well. Our model doesn't have a clear bias towards the base classes in the training set and could tell the difference very well that have no chance to learn in the COCO training: e.g toy vs real and filled vs empty.

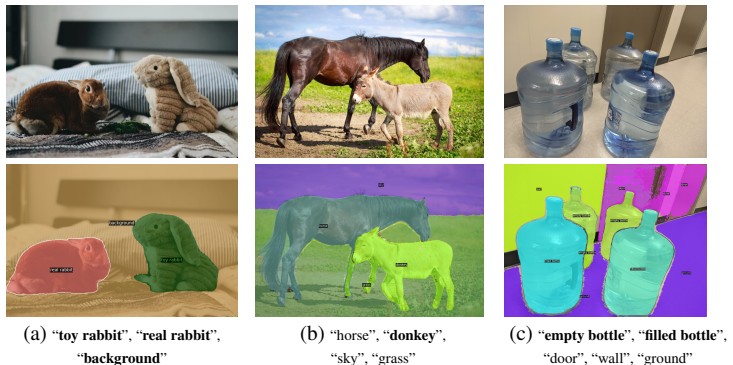

(a) "**toy rabbit**", "**real rabbit**", "**background**"

(b) "horse", "**donkey**", "sky", "grass"

(c) "**empty bottle**", "**filled bottle**", "door", "wall", "ground"

Figure 4: **User-specified class panoptic segmentation.** The labels above are the text inputs we used for testing the images. Texts in bold are novel classes i.e. don't exist in the labels of COCO training data. (a) Our model is able to distinguish object properties of real rabbit and toy rabbit. (b) This example shows that our model is potential for fine-grained classifications and does not have bias toward the base classes. (c) Our results show that it can tell the difference between the filled status and empty status of bottles.

## 4.4 OPEN-VOCABULARY SEMANTIC SEGMENTATION/LABELING

We also use our method to compare with open-vocabulary semantic segmentation as in Table 3. The setting is similar, they all train on COCO panoptic training set and test on ADE20K validation set. On the four datasets we test, MaskCLIP reaches the state-of-the-art results on three of them with only P-59 slightly lower.

Table 3: **Results on open-vocabulary semantic segmentation.** A-150 and A-847 represent the ADE20K dataset with 150 classes and 847 classes respectively. P-459 and P-59 represents PASCAL Context dataset with 459 classes and 59 classes respectively. All results use the mIoU metric. All methods presented here don't use extra data other than COCO for training.

| Method | COCO Training Data | A-150 ↑ | A-847 ↑ | P-459 ↑ | P-59 ↑ |
|---|---|---|---|---|---|
| ALIGN (Jia et al., 2021) | None | 10.7 | 4.1 | 3.7 | 15.7 |
| ALIGN w/ proposals (Jia et al., 2021) | Masks | 12.9 | 5.8 | 4.8 | 22.4 |
| LSeg+ (Li et al., 2022a) | Masks + Labels | 18.0 | 3.8 | 7.8 | **46.5** |
| OpenSeg (Ghiasi et al., 2022) | Masks + Captions | 21.1 | 6.3 | 9.0 | 42.1 |
| CLIP Baseline | Masks | 13.8 | 5.2 | 5.2 | 25.3 |
| MaskCLIP w/o RMA | Masks | 14.9 | 5.6 | 5.3 | 26.1 |
| MaskCLIP (MaskRCNN) | Masks + Labels | 22.4 | 6.8 | 9.1 | 41.3 |
| MaskCLIP | Masks + Labels | **23.7** | **8.2** | **10.0** | 45.9 |

To compare with previous methods, we also provide a semantic segmentation comparison in Figure 5. Results on ALIGN++ and OpenSeg are directly from (Ghiasi et al., 2022) and we run the same image using our MaskCLIP model. It can be seen that due to the open vocabulary setting, some similar classes may be mistakenly classified e.g. all three methods predict the house in this image while the ground truth is building.

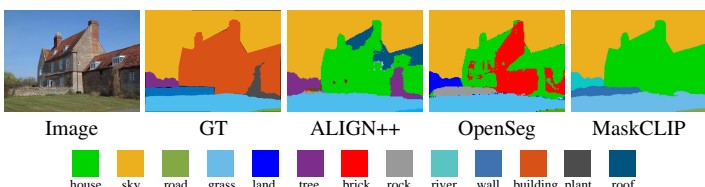

Image    GT    ALIGN++    OpenSeg    MaskCLIP

house    sky    road    grass    land    tree    brick    rock    river    wall    building    plant    roof

Figure 5: Comparison on open-vocabulary semantic segmentation. The input image and the results for GT, ALIGN++, OpenSeg are from (Ghiasi et al., 2022).

## 4.5 OPEN-VOCABULARY INSTANCE SEGMENTATION

**Cross-Dataset Setting.** We present the results on open vocabulary instance segmentation in Table 4 under the cross-dataset setting. Since instance segmentation can be regarded as "thing-only" panoptic segmentation, we directly apply our model trained on coco panoptic dataset to the instance segmentation task. MaskCLIP with different class-agnostic mask proposal networks perform better than CLIP Baseline and MaskCLIP w/o RMA in general.

Table 4: **Results on open-vocabulary instance segmentation using the ADE20k validation dataset and the LVIS validation dateset under the cross-dataset setting.**

| Method | ADE20K | | | LVIS | | |
|---|---|---|---|---|---|---|
| | AP ↑ | AP$^{50}$ ↑ | AP$^{75}$ ↑ | AP ↑ | AP$^{50}$ ↑ | AP$^{75}$ ↑ |
| CLIP Baseline | 3.974 | 6.090 | 4.288 | 4.989 | 7.244 | 5.227 |
| MaskCLIP w/o RMA | 4.263 | 6.696 | 4.402 | 5.762 | 8.202 | 6.169 |
| MaskCLIP (MaskRCNN) | **6.164** | **12.072** | 5.775 | 6.431 | **12.753** | 5.777 |
| MaskCLIP | 5.989 | 9.739 | **6.209** | **8.404** | 12.190 | **8.810** |

**COCO Split Setting.** Besides the cross-dataset setting, we also follow the COCO Split Setting in XPM(Huynh et al., 2022) to perform the instance segmentation in Table 5. On the generalized setting which is a more challenging setting, we outperforms previous results in base, target and all categories. On the constrained setting, we also show competitive results in both base and target categories.

Table 5: **Results on open-vocabulary instance segmentation under the COCO split setting.**

| Method | Constrained | | Generalized | | |
|---|---|---|---|---|---|
| | Base | Target | Base | Target | All |
| Soft-Teacher(Xu et al., 2021) | 41.8 | 14.8 | 41.5 | 9.6 | 33.2 |
| Unbiased-Teacher(Liu et al., 2021) | 41.8 | 15.1 | 41.4 | 9.8 | 33.1 |
| XPM(Huynh et al., 2022) | 42.4 | **24.0** | 41.5 | 21.6 | 36.3 |
| MaskCLIP | **42.8** | 23.2 | **42.6** | **21.7** | **37.2** |

## 5 ABLATION STUDY

**Incorporating GT Masks.** Since our model can decouple the mask proposal process and the classification process, we could also use the ground truth mask proposals which can be regarded as a "perfect" mask proposal network in our method. In this way, we can eliminate the effects of quality of the mask proposals and inspect the method's classification capabilities. In Table 6. We can see that the performance could gain a lot from the "perfect" mask proposals. And our MaskCLIP method also outperforms OpenSeg in this setting.

Table 6: **Incorporating GT Masks.** Results on using GT masks as mask proposals for open-vocabulary panoptic segmentation and semantic segmentation.

| | PQ ↑ | mIoU ↑ |
|---|---|---|
| OpenSeg (Ghiasi et al., 2022) | - | 21.1 |
| MaskCLIP | 15.1 | 23.7 |
| OpenSeg + GT masks (Ghiasi et al., 2022) | - | 27.5 |
| MaskCLIP + GT masks | 35.8 | 31.7 |

**Mask Refinement.** In our Relative Mask Attention part, the attention layer will use the accurate mask information to learn a better attention matrix and the mask will also use the attention information to gradually refine itself. In this ablation study, we only let the attention matrix learn from the mask without any mask refinement. And we get the results in Table 7. Since the SQ reflects the segemtation quality, we care more about SQ here. It can be seen that MaskCLIP performs slightly better than that without the mask refinement which demonstrates the effectivity of the mask refinement.

Table 7: **Ablation Study on Mask Refinement.** Results on ADE20K validation set are reported here. Both methods are trained on COCO and tested on ADE20K validation dataset.

| | PQ ↑ | PQ$^{Th}$ ↑ | PQ$^{St}$ ↑ | SQ ↑ | SQ$^{Th}$ ↑ | SQ$^{St}$ ↑ |
|---|---|---|---|---|---|---|
| MaskCLIP w/o mask refinement | 13.624 | 13.253 | 14.368 | 66.361 | 67.715 | 63.653 |
| MaskCLIP | 15.121 | 13.536 | 18.290 | 70.479 | 70.021 | 71.396 |

## 6 CONCLUSION

In this paper, we have presented a new algorithm, MaskCLIP, to tackle an emerging computer vision task, open-vocabulary panoptic segmentation. MaskCLIP is a Transformer-based approach using mask queries with the ViT-based CLIP backbone to efficiently and effectively utilize pre-trained dense/local CLIP features. MaskCLIP consists of a Relative Mask Attention (RMA) module that is seamlessly integrated with a pre-trained CLIP. MaskCLIP is distinct compared with existing approaches in open-vocabulary semantic segmentation/object detection by building an integrated encoder module for segmentation mask refinement and image feature extraction with a pre-trained

CLIP image model. Encouraging experimental results on open-vocabulary panoptic/semantic/instance segmentation have been obtained.

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

## A CLIP BASELINE DETAILS

Here we provide more details on our CLIP Baseline. Given an RGB image $\mathcal{I} \in \mathbb{R}^{H \times W \times 3}$ with height $H$ and width $W$ and a list of category names with $C$ classes, we precompute the text embedding of the category names as $\mathcal{E} \in \mathbb{R}^{C \times D}$. The mask proposal network $f_m$ outputs $N$ masks $\mathcal{M} \in \mathbb{R}^{N \times H \times W}$. For each mask: the cropped image region is the element-wise product between the binary mask $\mathcal{M}_i$ and the image $\mathcal{I}$, i.e. $\mathcal{R}_i \in \mathbb{R}^{H \times W \times 3}$; the visual embedding $\mathcal{V}_i \in \mathbb{R}^D$ of the cropped region is computed by the visual encoder where $D$ is the hidden dimension; the final classification score $\mathcal{Y}_i \in \mathbb{R}^C$ is the softmax over the dot product between the visual embedding $\mathcal{V}_i$ and the text embedding $\mathcal{T}$. A formal algorithm is described as 1 and a visualization of this is shown as 6.

---

**Algorithm 1** CLIP Baseline

---

**Require:** Mask proposal network $f_m$, CLIP visual encoder $f_v$, CLIP text encoder $f_t$.
  Given an image $\mathcal{I} \in \mathbb{R}^{H \times W \times 3}$ and a list $\mathcal{T}$ containing $C$ category names.
  $\mathcal{E} = f_t(\mathcal{T})$.
  $\mathcal{M} = f_m(\mathcal{I})$.
  **for** $t = 1, 2, \ldots, N$ **do**
    $\mathcal{R}_i = \mathcal{M}_i \odot \mathcal{I}$.
    $\mathcal{V}_i = f_v(\mathcal{R}_i)$.
    $\mathcal{Y}_i = \text{softmax}(\mathcal{E} \otimes \mathcal{V}_i)$.
  **end for**

---

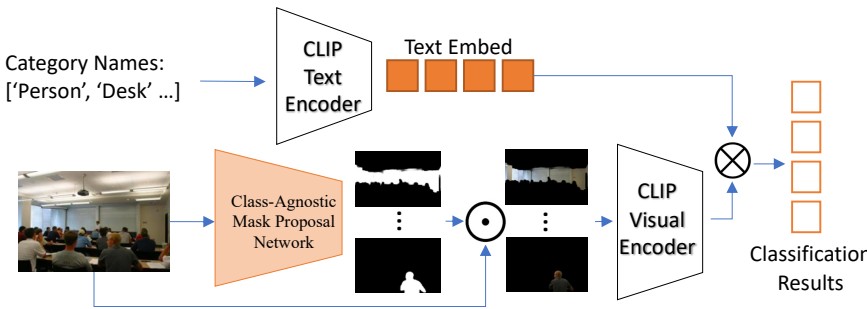

Figure 6: Illustration of the CLIP baseline.

## B ABLATION ON USING RELATIVE MASK ATTENTION IN DIFFERENT LAYERS

In this part we conduct an ablation study on using different layers for relative mask attention. Since our pretrained CLIP model is fixed during the whole training procedure, whether each layer would help the final results remains a question. We use four different kinds of layers combination of the layers in this part and provide the results in Table 8. We can see that the last layer is a key part of our results since the features are gradually learned through all the attention layer. Though the last four layers' features should the best, the performance wouldn't be better if Relative Mask Attention is only used in the last four layers. This is also reasonable since the network should not have the accurate mask information too late.

## C MORE VISUALIZATION RESULTS ON ARBITRARY CATEGORIES

In this part, we provide more visualization results on user-specified class discoveries in Figure 7. We select some very close text prompts such as "four-leg animal" and "two-leg animal"; "car", "truck" and "SUV" and find that our method can still classify them. We also show another result which is "person identification" in Figure 7 (c) which shows our model preserve the dense/local CLIP features rather well.

Table 8: **Ablation Study on Relative Mask Attention Layers in different layers.** All the methods are trained on COCO and tested on ADE20K validation dataset. The pretrained CLIP ViT-L/14@336px model has 24 layers and we replace four of them with our relative mask attention to fully utilize the accurate mask information and refine the masks.

| Different Layers | PQ | $PQ^{Th}$ | $PQ^{St}$ |
|---|---|---|---|
| 1, 7, 13, 19 | 11.241 | 10.519 | 12.686 |
| 3, 9, 15, 21 | 11.372 | 10.141 | 13.835 |
| 21, 22, 23, 24 | 14.673 | 14.048 | 15.922 |
| 6, 12, 18, 24 | 15.121 | 13.536 | 18.290 |

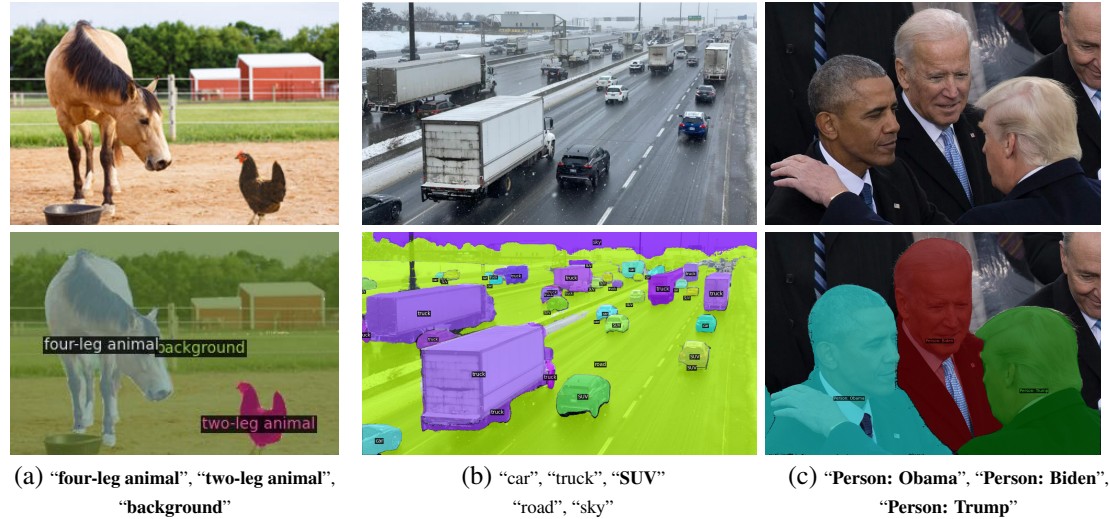

(a) "**four-leg animal**", "**two-leg animal**", "**background**"

(b) "car", "truck", "**SUV**" "road", "sky"

(c) "**Person: Obama**", "**Person: Biden**", "**Person: Trump**"

Figure 7: **More qualitative restuls on user-specified class.** The labels above are the text prompts we used for testing the images. Texts in bold are novel classes i.e. don't exist in the labels of COCO training data.

# D    QUALITATIVE RESULTS ON ADE-20K INSTANCE SEGMENTATION

We also show qualitative results on ADE20K comparing the CLIP Baseline, MaskCLIP w/o RMA and MaskCLIP. As shown in Figure 8, MaskCLIP is much better than the CLIP Baseline and MaskCLIP w/o RMA. Since the mask proposal network is class-agnostic, some masks that are not objects would actually be predicted in which case the classification later will be very important as it may be classified as some object classes in the dataset. The visualization results of MaskCLIP contains fewer non-object masks and are more accurate in class prediction.

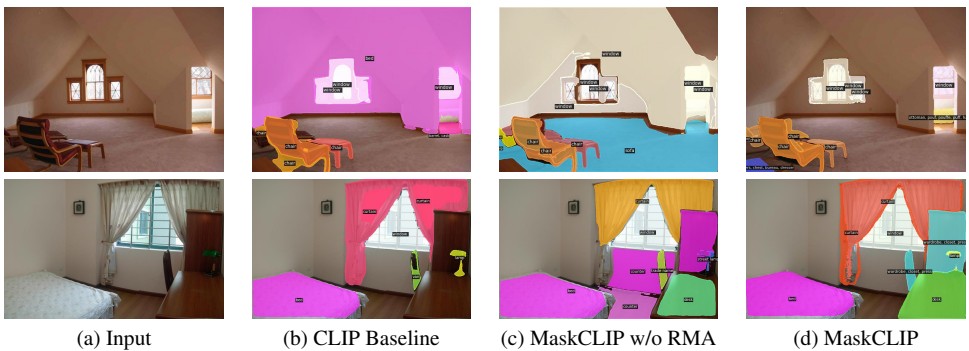

|     (a) Input     |    (b) CLIP Baseline    |    (c) MaskCLIP w/o RMA    |    (d) MaskCLIP    |

Figure 8: Qualitative results on ADE20K instance segmentation.

# E    COCO VALIDATION RESULTS

In this part, we provide the COCO validation results. The model is trained on COCO panotpic training data and evaluated on COCO validation data. This results provide more comparison on model's

Table 9: **Results on COCO validation dataset.** Panoptic and semantic segmentation tasks are both evaluated here which is not under the open vocabulary setting since all the classes are base classes. Results of ALIGN, ALIGN++ and OpenSeg are cited from Ghiasi et al. (2022). RMA refers to the Relative Mask Attention (RMA) module shown in Figure 1.

|  | Panoptic | | | Semantic |
|---|---|---|---|---|
|  | PQ | PQ$^{Th}$ | PQ$^{St}$ | mIoU |
| ALIGNJia et al. (2021) |  |  |  | 15.6 |
| ALIGN w/ proposalJia et al. (2021) |  |  |  | 17.9 |
| LSeg+Li et al. (2022a) |  | - |  | **55.1** |
| OpenSegGhiasi et al. (2022) |  |  |  | 36.1 |
| CLIP Baseline | 17.47 | 23.47 | 8.42 | 23.3 |
| MaskCLIP w/o RMA | 15.32 | 20.12 | 8.07 | 19.5 |
| MaskCLIP | **30.89** | **34.78** | **25.02** | 47.6 |

## F  EFFICIENCY ANALYSIS.

We also provide the FLOPs for differnt models in Table 10. We use an input resolution of 640x640 here. The CLIP visual encoder we use is ViT-L/14@336px. As we can see, our MaskCLIP w/o RMA and MaskCLIP's FLOPs are much lower than the CLIP Baseline which needs to send each image to the CLIP Visual Encoder.

Table 10: **FLOPs for differnt models.**

| Method | FLOPs |
|---|---|
| Mask2Former | 79G |
| CLIP Visual Encoder | 233G |
| CLIP Baseline | 23382G |
| MaskCLIP w/o RMA | 352G |
| MsakCLIP w/ RMA | 542G |

