# OpenReview forum: "Open-Vocabulary Panoptic Segmentation MaskCLIP"
_ICLR.cc/2023/Conference — Submitted to ICLR 2023_

### Official Review · Reviewer_MZxt · 2022-10-23

**Confidence:** 5
**Correctness:** 3
**Technical Novelty And Significance:** 3
**Empirical Novelty And Significance:** 3
**Recommendation:** 5

**Clarity, Quality, Novelty And Reproducibility:**

**Clarity**: a little weak, some details are missing and contributions are not well presented.

**Quality**: this paper is good work, at least the Mask Class Tokens. Not sure about the Relative Mask Attention (RMA), I hope the authors could address my concerns on RMA.

**Novelty**: the idea of Mask Class Tokens is novel.

**Reproducibility**: it's easy to reproduce the idea of mask class tokens.

**Strength And Weaknesses:**

**Strength**:

1) The idea of Mask Class Tokens is interesting and useful. Previous works usually crop the detected mask regions and then send the cropped regions to CLIP image encoder, which means the CLIP image encoder is called many times and processes the patches of the same image many times. With Mask Class Tokens, the CLIP could deal with the mask regions of the same image in parallel in a single forward. And experiments prove its efficiency.
2) The experiments show the proposed MaskCLIP achieves good results on open-vocabulary panoptic/semantic segmentation.

**Weaknesses**:

1) The paper is not well presented. It is a little hard to understand the ideas, especially Eq. (1). A detailed explanation of the four parts in Eq. (1) may help the reader to better understand.
2) Some key details are missing. What if $f_2$ in Figure 2? The text says "$f_1$ and $f_2$ are two downsampling networks...sharing the same architecture", does this mean $f_1$ is ViT also?
3) If yes to the above question, so the image-encoder of CLIP is called #mask times, and the forward process time should be the same as sending mask regions to CLIP one by one (or in a batch), which is not efficient.
4) Could you please provide the inference speed, like GFLOP, of w/ and w/o RMA, w/ and w/o Mask Class Tokens?
5) The proposed approach is **not real open-vocabulary**. The claimed "open-vocabulary" is based on Mask2former training data, so the mask2former can only propose masks like in training data. For example, the is class **Person** in COCO but no class **Leg**, so is the target-of-the-interest is **Leg**, the mask2former cannot provide a mask for **Leg**, so your approach should also fail in labeling **Leg**.
6) Your approach is actually unified to semantic/instance/panoptic segmentation, why emphasize panoptic in the title and abstract?
7) **[Training data]** Do you use all the categories of COCO panoptic segmentation? Do you set some of them as novel categories?

**Summary Of The Paper:**

The paper proposes a Transformer-based approach MaskCLIP to address open-vocabulary panoptic segmentation. It uses the existing Mask2former to propose Masks of objects/stuff, and label these detected Mask regions by sending them into CLIP. Instead of sending these mask regions one by one, it aims to modify the image-encoder (ViT in this paper) of CLIP to deal with the mask regions of the same image in parallel. To achieve this, Mask Class Tokens that are copied from class token are proposed to represent corresponding mask regions, the masks proposed by mask2former is used to mask the attention matrix in cross-attention, by which each mask class token focuses on a specific mask region. Then the paper proposes a Relative Mask Attention to 1) update the attention matrix of the mask class token and 2) refine segmentation masks. It achieves good performance on open-vocabulary panoptic/semantic segmentation.

**Summary Of The Review:**

Overall, the proposed Mask Class Token is interesting and useful, while the proposed Relative Mask Attention missing essential details and remains some concerns. The performance on open-vocabulary panoptic/semantic segmentation is good. I give a borderline score and am willing to raise the score if authors demonstrate more significance of the RAM.

---

> ### Author Response · Authors · 2022-11-16
> **Response to Reviewer MZxt**
>
> We thank reviewer MZxt for the valuable feedback.
>
> > The paper is not well presented. It is a little hard to understand the ideas, especially Eq. (1). A detailed explanation of the four parts in Eq. (1) may help the reader to better understand.
>
> **Answer:** Thanks for the suggestion. We have updated our manuscript accordingly. Please see the revised one in Mask Class Tokens where we explain more details on Eq 1.
>
> > Some key details are missing. What is $f_1$ in Figure 2? The text says " $f_1$ and  $f_2$ are two downsampling networks...sharing the same architecture", does this mean  is ViT also?
>
> **Answer:** No, f1 and f2 are just two downsampling convolution layers. More specifically, in our case, it’s conv2d(in_channels, out_channels, stride=patch_size, kernel_size=patch_size).
>
> > If yes to the above question, so the image-encoder of CLIP is called #mask times, and the forward process time should be the same as sending mask regions to CLIP one by one (or in a batch), which is not efficient.
>
> **Answer:** As f2 is not a ViT and only a conv2d, we don’t need to call image-encoder of CLIP #mask times.
>
> > Could you please provide the inference speed, like GFLOP, of w/ and w/o RMA, w/ and w/o Mask Class Tokens?
>
> **Answer:**  Thanks for the suggestion. We put the table here while also reported the FLOPs in the common reply above and updated the manuscript in Table 10.
>
> | Model | FLOPs |
> | ----- | -----------------------------------------------------|
> | Mask2Former | 79 GFLOPs |
> | CLIP Visual Encoder | 233 GFLOPs |
> | CLIP Baseline | 23382 GFLOPs |
> | MaskCLIP w/o RMA  | 352 GFLOPs |
> | MsakCLIP w/ RMA | 542 GFLOPs |
>
> > The proposed approach is not real open-vocabulary. The claimed "open-vocabulary" is based on Mask2former training data, so the mask2former can only propose masks like in training data. For example, the is class Person in COCO but no class Leg, so is the target-of-the-interest is Leg, the mask2former cannot provide a mask for Leg, so your approach should also fail in labeling Leg.
>
> **Answer:** Thanks for raising this point. We find that though our training data is only for some specific classes, our class-agnostic proposal network could still output masks that belong to the novel classes mainly because since there is no class information provided, the network is learning generic mask characteristics. But yes, the class-agnostic proposal network could not generate all the masks in the image though, like legs, eyes, faces etc. Previous works like OpenSeg[1] also utilizes a class-agnostic mask proposal network. We believe this is a common problem here. To further solve it, one might need to incorporate all the novel class information in the mask-proposing process which would be a good future direction.
>
> > Your approach is actually unified to semantic/instance/panoptic segmentation, why emphasize panoptic in the title and abstract?
>
> **Answer:** Thank you for raising this point. Please see our common reply on why we titled our paper panoptic segmentation. We've changed the title to panoptic/universal to address the concern.
>
> > [Training data] Do you use all the categories of COCO panoptic segmentation? Do you set some of them as novel categories?
>
> **Answer:** Yes. For experiments in the paper, we follow OpenSeg[1] to use COCO panoptic as training data. During testing, we use ADE20k which contains novel categories that don’t exist in COCO. Additionally, we follow XPM[2] under the split settings including 48 base classes for training and 17 novel classes for evaluation. We report our results in the common reply above and also updated the manuscript in Table 5.
>
> > ...am willing to raise the score if authors demonstrate more significance of the RMA.
>
> **Answer:**  Please see the common reply above on the significance of RMA where we discuss the significance of the RMA module.
>
> [1]  [Scaling Open-Vocabulary Image Segmentation with Image-Level Labels](https://arxiv.org/pdf/2112.12143). Golnaz Ghiasi, Xiuye Gu, Yin Cui, Tsung-Yi Lin. ECCV 2022\
> [2]  [Open-Vocabulary Instance Segmentation via Robust Cross-Modal Pseudo-Labeling](https://openaccess.thecvf.com/content/CVPR2022/papers/Huynh_Open-Vocabulary_Instance_Segmentation_via_Robust_Cross-Modal_Pseudo-Labeling_CVPR_2022_paper.pdf). Dat Huynh, Jason Kuen, Zhe Lin, Jiuxiang Gu and Ehsan Elhamifar. CVPR 2022

---

### Official Review · Reviewer_GGkA · 2022-10-25

**Confidence:** 4
**Correctness:** 4
**Technical Novelty And Significance:** 3
**Empirical Novelty And Significance:** 3
**Recommendation:** 6

**Clarity, Quality, Novelty And Reproducibility:**

The paper is well-written, in good quality, and novel. It's currently hard to reproduce without code due to the complexity of the system.

**Strength And Weaknesses:**

Strength

+ The task of open-vocabulary segmentation is important, and the authors proposed a reasonable (and unified) framework for open-vocabulary segmentation panoptic, semantic, and instance.

+ I appreciate that the evaluation is done in an cross-dataset setting, which can justify the generalization ability of the proposed framework.

+ The proposed mask attention module gives a good improvements in Table 2, Table 3, and Table 4.

+ The overall framework outperforms strong state-of-the-art semantic segmentation methods including LSeg, OpenSeg, and ALIGN as shown in Table 3.

Weaknesses

- While the paper aims at open-vocabulary segmentation, I feel the idea of using a mask attention module is not directly related to open-vocabulary --- it seems that this module could also improve standard segmentation, with the use of the CLIP visual encoder as an feature refiner. Please correct me if I am wrong, otherwise it would be better to also provide experiments on close-vocabulary segmentation.

- The overall framework looks computationally expensive. It adds an additional ViT-L backbone from CLIP visual encoder, after an already-expensive Mask2Former module. It will be nice if the authors can discuss the FLOPs or latency for better understanding the trade-off.

- Except for semantic segmentation experiments, the authors did not compare to other papers, and the numbers compared to the close-vocabulary setting is low. This makes these two settings not calibrated and less convincing. I understand that there are no/ few works reporting numbers on this specific setting, both some other comparable settings might be viable. E.g., the LVIS base/ novel split in ViLD.



**Summary Of The Paper:**

This paper works on open-vocabulary panoptic segmentation using pretrained CLIP weights. The authors first propose a new panoptic segmentation framework that first do class-agnostic mask proposal using an existing frameworks (MaskRCNN and Mask2Former), and then use the predicted mask to interact with the CLIP visual backbone via a relative mask attention module (RMA). The final classification prediction (for any classes) is the dot-product between the refined mask features and the fixed CLIP language embedding. Experiments show the proposed framework outperform existing semantic segmentation methods on ADE20K and Pascal with a good margin (Table. 3), and the proposed mask attention module played an important role in the improvements.

**Summary Of The Review:**

This paper propose a valid framework for an important and challenging problem. The author shows state-of-the-art performance on one of the benchmarks (semantic segmentation), but is not super convincing on others (panoptic and instance segmentation). There are also some concerns on the complexity of the added component. My current rating is a borderline accept (mostly for the cross-dataset evaluation setup), but might change (increase or decrease) after rebuttal.

---

> ### Author Response · Authors · 2022-11-16
> **Response to Reviewer GGkA**
>
> We thank reviewer GGkA for the valuable feedback.
>
> > While the paper aims at open-vocabulary segmentation, I feel the idea of using a mask attention module is not directly related to open-vocabulary --- it seems that this module could also improve standard segmentation, with the use of the CLIP visual encoder as an feature refiner. Please correct me if I am wrong, otherwise it would be better to also provide experiments on close-vocabulary segmentation.
>
> **Answer:** Yes, that’s true, the mask attention module is not directly related to open-vocabulary. Deformable Detr uses a similar mechanism in their method which is for closed-vocabulary detection. In ours, a key difference is that the feature extractor CLIP is frozen, our mask attention module is to compute the partial/dense features of the image more efficiently given the specific masks.
>
> > The overall framework looks computationally expensive. It adds an additional ViT-L backbone from CLIP visual encoder, after an already-expensive Mask2Former module. It will be nice if the authors can discuss the FLOPs or latency for better understanding the trade-off.
>
> **Answer:** Thanks for the suggestion, we have reported the FLOPs for different methods in the common reply above and also updated the manuscript in Table 10.
>
> > Except for semantic segmentation experiments, the authors did not compare to other papers, and the numbers compared to the close-vocabulary setting is low. This makes these two settings not calibrated and less convincing. I understand that there are no/ few works reporting numbers on this specific setting, both some other comparable settings might be viable. E.g., the LVIS base/ novel split in ViLD.
>
> **Answer:** Thanks for the suggestion, we have followed the COCO split setting in XPM[1] and report our results in the common reply and also updated the manuscript in Table 5.
>
> [1]  [Open-Vocabulary Instance Segmentation via Robust Cross-Modal Pseudo-Labeling](https://openaccess.thecvf.com/content/CVPR2022/papers/Huynh_Open-Vocabulary_Instance_Segmentation_via_Robust_Cross-Modal_Pseudo-Labeling_CVPR_2022_paper.pdf). Dat Huynh, Jason Kuen, Zhe Lin, Jiuxiang Gu and Ehsan Elhamifar. CVPR 2022

---

> > ### Comment · Reviewer_GGkA · 2022-11-30
> > **Thank you for your response**
> >
> > I appreciate the efforts the authors put in the rebuttal. Most of my concerns are NOT addressed in the rebuttal. Given that all other reviewers vote for a reject, I do not feel it is strong enough for me to champion this paper, and I am OK if this paper get rejected. Pleas find my detailed response to each point below.
> >
> > > Connection between Mask-attention module and Open-vocabulary.
> >
> > The authors agreed these two ingredients are not fully connected. This makes the story a bit un-appealing. If the paper is proposing a general module that helps segmentation, only evaluating it on open-vocabulary is not satisfactory.
> >
> > > Computation.
> >
> > I appreciate the authors providing the FLOPs for reference. However these numbers are not positive and could not be a reason for me to improve my rating.
> >
> > > Compare to other methods.
> >
> > Thank the authors for providing the numbers. The results are good but not strong (epsilon better than existing work). Also, I am still preferring LVIS settings for open-vocabulary evaluation. I am less convinced that we can learn a general semantic model on <100 classes.

---

> > > ### Author Response · Authors · 2022-12-01
> > > **Response to Reviewer GGkA**
> > >
> > > We thank reviewer GGkA for the reply!
> > >
> > > > The authors agreed these two ingredients are not fully connected. This makes the story a bit un-appealing. If the paper is proposing a general module that helps segmentation, only evaluating it on open-vocabulary is not satisfactory.
> > >
> > > Sorry if our initial response is causing some misunderstandings here. The mask attention module can be used for closed vocabulary settings which has been explored in Deformable-DETR where during training an attention mask is applied in the transformer. However, the key difference is that our transformer(CLIP visual encoder) is pretrained without mask module attention and fixed. We add the mask attention module on a fixed pretrained transformer network and obtain the dense/clip features. The mechanism itself is not restricted to open-vocabulary, but the usage of adding the mask attention module to a fixed transformer is only applied on the open-vocabulary setting given the pretrained CLIP visual encoder. Our work tries to design a simple yet effective way to extract the dense/partial features of the CLIP(which is trained for extracting the global image features) so that no cropping or distillation is needed for extracting dense/partial features.
> > >
> > > > I appreciate the authors providing the FLOPs for reference. However these numbers are not positive and could not be a reason for me to improve my rating.
> > >
> > > The RMA itself brings some more FLOPs as it contains parameters and computation.  However, many previous methods using distillation requires cropping features. Their computational complexities are close to our CLIP Baseline's which is extremely expensive as shown above in the common reply. For example, suppose there are 100 cropped regions/masked images, that means one needs to run CLIP for 100 times. Thus the total FLOPs would be 233Gx100=23.3T. The point is we want to show that the RMA design does not bring too many FLOPs compared to the MaskCLIP w/o RMA and the whole model reduces the computation cost by a lot compared to directly cropping the images and getting the dense/partial features.
> > >
> > > > Thank the authors for providing the numbers. The results are good but not strong (epsilon better than existing work). Also, I am still preferring LVIS settings for open-vocabulary evaluation. I am less convinced that we can learn a general semantic model on <100 classes.
> > >
> > > Since ours is a generic method for semantic/instance/panoptic segmentation and not a dedicated open-vocabulary instance segmentation work, we evaluate the COCO split setting and not LVIS split setting following Mask2former. LVIS, on the other hand, requires a more dedicated instance-segmentation method as it contains more instances and classes. For example, two current open-vocabulary instance segmentation works that perform on LVIS split setting, VilD[1] and OV-DETR[2] are actually two open-vocabulary object detection frameworks which add an external mask head for instance segmentation.
> > >
> > > [1] [Open-vocabulary object detection via vision and language knowledge distillation](https://openreview.net/pdf?id=lL3lnMbR4WU). Xiuye Gu, Tsung-Yi Lin, Weicheng Kuo and Yin Cui. ICLR 2022\
> > > [2] [Open-Vocabulary DETR with Conditional Matching](https://arxiv.org/pdf/2203.11876.pdf). Yuhang Zang, Wei Li, Kaiyang Zhou, Chen Huang, Chen Change Loy. ECCV 2022

---

### Official Review · Reviewer_TB5t · 2022-10-25

**Confidence:** 5
**Correctness:** 3
**Technical Novelty And Significance:** 2
**Empirical Novelty And Significance:** 2
**Recommendation:** 5

**Clarity, Quality, Novelty And Reproducibility:**

The paper is clearly written and easy to follow. The source code is not provided.

**Strength And Weaknesses:**

Strength:

+ It is reasonable to adapte CLIP module to better take care of mask inputs by applying and learning a mask attention module. The idea makes sense to me.

+ The proposed method shows improvements over the baseline model on both panoptic segmentation and semantic segmentation tasks. It also achieves better performance than some existing open-vocabulary semantic segmentation methods.

+ Ablations studies and analysis are conducted to help understand contribution of each part of the model.


Concerns:

- The key component of the proposed method is RMA module, which is a kind of general idea for adapting CLIP for Open-Vocabulary Panoptic Segmentation. There is no specific designs for panoptic segmentation. Therefore, it is not clear to me why authors emphasize panoptic segmentation where there is no existing methods for direction comparison.

-  For panoptic segmentation, in addition to very basic CLIP baseline, it would be more convincing to benchmark SOTA Open-Vocabulary Segmentation method, e.g. OpenSeg or others by using the same level of supervision to enable comparisons for panoptic segmentation.

-  Authors claimed efficiently for the proposed method and strength than avoids the student-teacher training process. However, evidences are not sufficient to support such claim. For example, there is no inference time comparison between the proposed method and existing method in Table 3. There is also no experiments to compare the proposed method and student-teacher training methods, e.g. [1]. Does the proposed method running faster than [1] or achieve better performance than [1]?

[1] Open-vocabulary object detection via vision and language knowledge distillation, ICLR 2022

- Are comparisons in Table 3 fair? the proposed method rely on both semantic and instance level annotations (panoptic segmentation) to conduct training. Is this the same case for those Open-Vocabulary Semantic Segmentation models?

- Authors mention many time that the proposed method does not need finetuning. Does the proposed method perform better than finetuning CLIP on masked data?

**Summary Of The Paper:**

This paper focuses on Open-Vocabulary Panoptic Segmentation where the goal is to conduct panoptic segmentation for arbitrary categories classes (open-vocabulary). A method called MaskCLIP is proposed to build on top of canonical background and instance mask representation with a cascade mask proposal and refinement process. Specifically, a Relative Mask Attention (RMA) module is presented on top of a ViT CLIP model to integrate mask tokens for semantic and instance level segmentations. Experiments are conducted on both open-vocabulary panoptic segmentation and semantic segmentation tasks including ADE20K and PASCAL datasets, which shows improvements over the baseline model or some existing methods.

**Summary Of The Review:**

The proposed method has encouraging results for Open-Vocabulary Semantic Segmentation. However, many major claims are not well supported by the current experiments. Please refer to the Strength And Weaknesses part for more details.

---

> ### Author Response · Authors · 2022-11-16
> **Response to Reviewer TB5t**
>
> We thank reviewer TB5t for the valuable feedback.
>
> > The key component of the proposed method is RMA module, which is a kind of general idea for adapting CLIP for Open-Vocabulary Panoptic Segmentation. There is no specific designs for panoptic segmentation. Therefore, it is not clear to me why authors emphasize panoptic segmentation where there is no existing methods for direction comparison.
>
> **Answer:** Thank you for raising this point. Please see our common reply above for why we titled panoptic. We've changed the title to panoptic/universal to address the concern.
>
> > For panoptic segmentation, in addition to very basic CLIP baseline, it would be more convincing to benchmark SOTA Open-Vocabulary Segmentation method, e.g. OpenSeg or others by using the same level of supervision to enable comparisons for panoptic segmentation.
>
> **Answer:** Since neither OpenSeg[3] nor ALIGN[4] releases their code, we are unable to benchmark their performance on panoptic segmentation. Thus we report both the semantic results and instance results in our paper which can be seen as the ability to do stuff and instance segmentation in panoptic segmentation. To further validate the effectiveness of our model, we follow experimental settings of XPM[2] and report our results on instance segmentation above in the common reply.
>
> > Authors claimed efficiently for the proposed method and strength than avoids the student-teacher training process. However, evidences are not sufficient to support such claim. For example, there is no inference time comparison between the proposed method and existing method in Table 3. There is also no experiments to compare the proposed method and student-teacher training methods, e.g. [1]. Does the proposed method running faster than [1] or achieve better performance than [1]?
>
> **Answer:** Thanks for the suggestion. We have reported the FLOPs in the common reply above and also updated the manuscript in Table 10. Ours and VilD[1] are not for the same task thus it’s hard to provide a direct comparison. However, for ViLD, it is similar to the CLIP baseline as it needs to send all the cropped regions to CLIP which is the same as in our CLIP baseline. Assume there are 100 cropped regions/masked images, that means one needs to run CLIP for 100 times. Thus the total FLOPs would be 233Gx100=23.3T.
>
> > Are comparisons in Table 3 fair? the proposed method rely on both semantic and instance level annotations (panoptic segmentation) to conduct training. Is this the same case for those Open-Vocabulary Semantic Segmentation models?
>
> **Answer:** The comparison is fair. OpenSeg[3] also uses panoptic annotations for training but only reports results on semantic segmentation.
>
> > Authors mention many time that the proposed method does not need finetuning. Does the proposed method perform better than finetuning CLIP on masked data?
>
> **Answer:** OpenSeg[3] finetunes ALIGN[4] which is similar to CLIP. As shown in Table3, our MaskCLIP achieves better results without fine-tuning. Furthermore, finetuning such a large model requires much training effort and also may destroy the learned feature space in the pretrained model.
>
> [1] [Open-vocabulary object detection via vision and language knowledge distillation](https://openreview.net/pdf?id=lL3lnMbR4WU). Xiuye Gu, Tsung-Yi Lin, Weicheng Kuo and Yin Cui. ICLR 2022\
> [2] [Open-Vocabulary Instance Segmentation via Robust Cross-Modal Pseudo-Labeling](https://openaccess.thecvf.com/content/CVPR2022/papers/Huynh_Open-Vocabulary_Instance_Segmentation_via_Robust_Cross-Modal_Pseudo-Labeling_CVPR_2022_paper.pdf). Dat Huynh, Jason Kuen, Zhe Lin, Jiuxiang Gu and Ehsan Elhamifar. CVPR 2022\
> [3] [Scaling Open-Vocabulary Image Segmentation with Image-Level Labels](https://arxiv.org/pdf/2112.12143). Golnaz Ghiasi, Xiuye Gu, Yin Cui, Tsung-Yi Lin. ECCV 2022\
> [4] [Scaling Up Visual and Vision-Language Representation Learning With Noisy Text Supervision](http://proceedings.mlr.press/v139/jia21b/jia21b.pdf). Chao Jia, Yinfei Yang, Ye Xia, Yi-Ting Chen, Zarana Parekh, Hieu Pham, Quoc V. Le,
> Yunhsuan Sung, Zhen Li and Tom Duerig. ICML 2022

---

### Official Review · Reviewer_a6DX · 2022-10-25

**Confidence:** 5
**Correctness:** 3
**Technical Novelty And Significance:** 2
**Empirical Novelty And Significance:** 2
**Recommendation:** 5

**Clarity, Quality, Novelty And Reproducibility:**

The motivation and the core idea of this paper are clear. However, it lacks many explanations for the details.
The novelty of this paper is limited. The core component RMA seems ok, but it brings many parameters and computations. However, the author does not prove it could beat simple alternatives.


**Strength And Weaknesses:**

**Strength**

The author builds a reasonable pipeline for open-vocabulary panoptic segmentation that could be trained in a data-efficient way.
The author designs RMA, which could effectively use the CLIP visual representation to support open-vocabulary recognition and mask refinement.
The qualitative and quantitative results are both competitive.


**Weaknesses**

**1)** Lack of experiments to support the claim

**a)** The performance of COCO is not reported. Getting a good trade-off between closed and open-vocabulary settings is also important.

**b)** RMA is the core contribution with a complicated structure. However, the author does not carry out ablation studies for the detailed structure of RMA. They could not prove the design of RMA is effective.

**c)** Only report the experiment results for one backbone, which is not convincing.

**d)** The title is panoptic segmentation, however, the experiment related to panoptic segmentation is only Table2 with 4 lines,  which is far from sufficient. The authors should at least include more naive baselines, like using different kinds of methods to fuse the CLIP representation.

**2)** Lack of explanations

**a)** How to convert the segmentation model to class-agnostic? If you just remove the class head, how to rank and remove duplicated masks?

**b)** What kind of Prompt is used is not mentioned, which is important.

**c)** The implementation of Mask Refinement is not mentioned.

**d)** The loss functions are not mentioned.

**e)** According to Section 4.2, the model is trained for only 10k iterations. However, the normal config on coco is 12 or 36 epochs, and Mask2former trains for 50 epochs.


**Summary Of The Paper:**

This paper proposes MaskCLIP, an open-vocabulary segmentation solution that could naturally deal with arbitrary categories using CLIP’s text representation. The newly proposed RMA module enables MaskCLIP to effectively leverage the prior knowledge contained in the CLIP visual encoder. Thus, only trained on COCO, MaskCLIP shows competitive performance on cross-dataset validation and open-vocabulary prediction.

**Summary Of The Review:**

The overall structure is clear, the writing needs further polishing with more explanations of details, and the contributions are overclaimed without sufficient support. The manuscript seems not well prepared and needs further investigation.

---

> ### Author Response · Authors · 2022-11-16
> **Response to Reviewer a6DX**
>
> We thank reviewer a6DX for the valuable feedback.
>
> > The performance of COCO is not reported. Getting a good trade-off between closed and open-vocabulary settings is also important.
>
> **Answer:** Thanks for the suggestion, we put the COCO validation results here and also put the COCO validation results in the revised version(Table 9).
>
> |Model  |Panoptic ||| Semantic|
> | --- | --- | --- | --- | --- |
> |  | PQ | PQ$^{\text{th}}$ | PQ$^{\text{st}}$ | mIoU |
> | ALIGN | | | | 15.6 |
> |ALIGN w/ proposal| | | | 17.9 |
> |LSeg+| | | | 55.1 |
> |OpenSeg| | | | 36.1 |
> |CLIP Baseline| 17.47 | 23.47 | 8.42 | 23.3 |
> |MaskCLIP w/o RMA |15.32| 20.12| 8.07| 19.5|
> |MaskCLIP |30.89 |34.78 |25.02 |47.6|
>
> > RMA is the core contribution with a complicated structure. However, the author does not carry out ablation studies for the detailed structure of RMA. They could not prove the design of RMA is effective.
>
> **Answer:** Both Mask Class Tokens and RMA are our core contributions. We have demonstrated our RMA’s significance in the common reply. We take ablation studies on removing the mask refinement part in Table 7 and we also explore different RMA layers’ design in Table 8.
>
> > Only report the experiment results for one backbone, which is not convincing.
>
> **Answer:** Thanks for the suggestion. For the class-agnostic mask proposal networks, we've already reported Mask2Former and MaskRCNN results in Table 2 and Table 3. We also ran another experiment by using ViT-L/14 of CLIP and we report the results here on panoptic segmentation and semantic segmentation.
>
> |Model  |Panoptic ||| Semantic|
> | --- | --- | --- | --- | --- |
> |  | PQ | PQ$^{\text{th}}$ | PQ$^{\text{st}}$ | mIoU |
> | OpenSeg | -|-| - | 21.1 |
> | Ours(ViT-L/14) | 14.014 |12.371 | 17.301 | 22.1 |
> | Ours(ViT-L/14@336px) | 15.121  |13.536| 18.290 | 23.7 |
>
> > The title is panoptic segmentation, however, the experiment related to panoptic segmentation is only Table2 with 4 lines, which is far from sufficient. The authors should at least include more naive baselines, like using different kinds of methods to fuse the CLIP representation.
>
> **Answer:** Thank you for raising this point. Please see our common reply above for why we titled panoptic. We've changed the title to panoptic/universal to address the concern. Since there are no direct methods for comparison (OpenSeg and ALIGN didn't release their codes), we compare our methods on both semantic segmentation and instance segmentation as they can represent "stuff" and "instance" segmentation in panoptic segmentation. Furthermore, we follow the COCO split setting in XPM[1] and report the results in the common reply.
>
> > How to convert the segmentation model to class-agnostic? If you just remove the class head, how to rank and remove duplicated masks?
>
> **Answer:** For MaskRCNN, we do the NMS according to the binary object score. For Mask2Former, it does not need to remove duplicated masks which are common practice in all transformer-based segmentation methods as they use matching loss during training.
>
> > What kind of Prompt is used is not mentioned, which is important.
>
> **Answer:** The textual inputs are simply “[class_name]”. We don’t further use any prompt engineering.
>
> > The implementation of Mask Refinement is not mentioned.
>
> **Answer:** We described our mask refinement process in Eq. 5. Basically it uses the attention matrix to compute a mask residual and add to the previous mask which shares the same idea as DETR to use attention matrix to compute the segmentation mask.
>
> > The loss functions are not mentioned.
>
> **Answer:** Thanks for the suggestion. We have updated our manuscript and we described our loss functions in the implementation details. It uses Hungarian matching to match the predicted masks and the gt masks and compute the classification and mask loss.
>
> > According to Section 4.2, the model is trained for only 10k iterations. However, the normal config on coco is 12 or 36 epochs, and Mask2former trains for 50 epochs.
>
> **Answer:** For the first stage, we train Mask2Former for 50 epochs as default. The second stage is more of a classification stage which CLIP already has the capability. That’s why our MaskCLIP w/o RMA baseline doesn’t need training at all and performs the segmentation. Our stage 2 training is just to let the RMA learn how to utilize the detailed mask information and refine the mask which doesn’t have many parameters. We find that 10k iterations is already enough for the network to converge during the experiments.
>
> [1] [Open-Vocabulary Instance Segmentation via Robust Cross-Modal Pseudo-Labeling](https://openaccess.thecvf.com/content/CVPR2022/papers/Huynh_Open-Vocabulary_Instance_Segmentation_via_Robust_Cross-Modal_Pseudo-Labeling_CVPR_2022_paper.pdf). Dat Huynh, Jason Kuen, Zhe Lin, Jiuxiang Gu and Ehsan Elhamifar. CVPR 2022

---

### Author Response · Authors · 2022-11-16
**Answers to Common Questions**

**1. Why titled panoptic?**

Panoptic segmentation unifies semantic segmentation (stuff or the background class like sky, road that don’t have the concept of individual instances) and instance segmentation (things like car, table, chair that are instance based). With the development of Transformer-based methods like Mask2Former where both stuff and things can be represented by general tokens, image segmentation tasks become increasingly “universal” and the distinction amongst semantic, instance, and panoptic segmentation becomes smaller. Nevertheless, the specific task of open-vocabulary panoptic segmentation has not been tackled before (to the best of our knowledge). In general, if one can do open-vocabulary panoptic segmentation, then open-vocabulary semantic and instance segmentations are sub-tasks that can be performed naturally. We have revised the paper title to “Open-Vocabulary Panoptic/Universal Segmentation with MaskCLIP" to address the concern.

**2. The significance of RMA**

RMA is an important part of our proposed MaskCLIP Visual Encoder. First we have shown different results with and without RMA in Table 2 and Table 3 on both panoptic setting and semantic setting and prove that the results with RMA can improve by over 50% on all the settings.
The reason why we design the RMA is: 1) **Provide detailed mask information** as the Mask Class Tokens can only utilize a low-resolution mask information. (Masks are actually the same resolution with the images (e.g. 336x336), however, the Mask Class Tokens can only use a much lower resolution mask, the attention mask it uses is just 24x24 in our case). So the RMA is significant to utilize the detailed mask information.
2) **Refine the mask.** As our class-agnostic mask proposal network is frozen during stage 2, the mask cannot be changed which is a limitation since it cannot utilize the supervision during stage 2.  We also show an ablation study in Table 6 to prove that our mask refinement is effective.

**3. Efficiency Analysis**

We provide the FLOPs here and also updated the manuscript in Table 10, we use 640x640 as input to compute the FLOPs, the CLIP Visual encoder we use is ViT-L/14@336px. As we can see below, the RMA did bring more computation to the model, but it still is way fewer than CLIP Baseline which is 23382 GFLOPs.
| Model | FLOPs |
| ----- | -----------------------------------------------------|
| Mask2Former | 79 GFLOPs |
| CLIP Visual Encoder | 233 GFLOPs |
| CLIP Baseline | 23382 GFLOPs |
| MaskCLIP w/o RMA  | 352 GFLOPs |
| MsakCLIP w/ RMA | 542 GFLOPs |

**4. COCO Split Comparison on Instance Segmentation**

To compare with more methods, we also follow the COCO split setting in XPM[1] to conduct instance segmentation under the same setting. We report our results here and also updated the manuscript in Table 5.

|Method| Constrained || Generalized |||
|--|--|--|--|--|--|
| | **Base** | **Target** | **Base** | **Target** | **All** |
| Soft-Teacher [2] | 41.8 | 14.8 | 41.5 | 9.6 | 33.2 |
| Unbiased-Teacher [3] | 41.8 | 15.1 | 41.4 | 9.8 | 33.1 |
| XPM [1] | 42.4 | **24.0** | 41.5 | 21.6 | 36.3 |
| Ours | **42.8** | 23.2 | **42.6** | **21.7** | **37.2** |

On the generalized setting which is a more challenging setting, we outperforms previous results in base, target and all categories. On the constrained setting, we also show competitive results in both base and target categories.


[1] [Open-Vocabulary Instance Segmentation via Robust Cross-Modal Pseudo-Labeling](https://openaccess.thecvf.com/content/CVPR2022/papers/Huynh_Open-Vocabulary_Instance_Segmentation_via_Robust_Cross-Modal_Pseudo-Labeling_CVPR_2022_paper.pdf). Dat Huynh, Jason Kuen, Zhe Lin, Jiuxiang Gu and Ehsan Elhamifar. CVPR 2022\
[2] [End-to-end semi-supervised object detection with soft teacher](https://openaccess.thecvf.com/content/ICCV2021/papers/Xu_End-to-End_Semi-Supervised_Object_Detection_With_Soft_Teacher_ICCV_2021_paper.pdf). Mengde Xu, Zheng Zhang, Han Hu,
Jianfeng Wang, Lijuan Wang, Fangyun Wei, Xiang Bai and Zicheng Liu. ICCV 2021\
[3] [Unbiased teacher for semisupervised object detection](https://arxiv.org/pdf/2102.09480.pdf). Yen-Cheng Liu, Chih-Yao Ma, Zijian He, Chia-Wen Kuo, Kan Chen, Peizhao Zhang, Bichen Wu, Zsolt Kira and Peter Vajda. ICLR 2021

---

### Decision · Program_Chairs · 2023-01-20

**Decision:**

Reject

**Justification For Why Not Higher Score:**

- All the reviewers raised concerns about the complexity of the work, the choice of only evaluating on panoptic segmentation, and the weak performance on closed-world classes.
- The author rebuttal does not address these three points well.
- Even the most positive reviewer (GGkA), finally admitted that they are not willing to champion the paper and are NOT excited about the work.


**Justification For Why Not Lower Score:**

N/A

**Metareview: Summary, Strengths And Weaknesses:**

*Summary*: The authors propose a framework for open-vocabulary panoptic segmentation that leverages a pre-trained vision-language model (CLIP). The framework consists of three stages - (1) predicting class agnostic masks (2) refining the mask feature by RMA (relative mask attention) interactions with the CLIP visual model and (3) using the CLIP language embeddings for open-vocabulary classification. The framework is evaluated on the ADE20K, Pascal and COCO (author comments) benchmarks.

*Strengths* - (1) As also noted by two reviewers, the proposed framework is generic enough without specific design choices for panoptic segmentation. (2) The RMA module proposed in this work is an interesting and empirically sound (Table 2/3) way to use the CLIP visual encoder. This is particularly useful in combination with Mask class tokens, since compared to prior work like ViLD, it does not require separate forward passes of each crop. (3) Empirical analysis on cross-dataset performance is quite good and demonstrates that the model is quite robust.

*Weaknesses*: (1) The method is evaluated only on panoptic segmentation benchmarks, rather than standard segmentation benchmarks. As noted earlier, the method does not have any panoptic specific ideas or design decisions. This seems to be purely an experimental design choice that is not addressed even after multiple reviewers bring it up. Lack of such experiments makes it hard to compare this work in a setting where prior open-vocabulary segmentation methods report their results. (2) The closed-world performance of this method is low, which makes it practically less useful. (3) The reviewers agree that the method is quite complex with marginal empirical gains.

**Summary Of Ac-Reviewer Meeting:**

- All the reviewers raised concerns about the complexity of the work, the choice of only evaluating on panoptic segmentation, and the weak performance on closed-world classes.
- The author rebuttal does not address these three points well.
- Even the most positive reviewer (GGkA), finally admitted that they are not willing to champion the paper and are NOT excited about the work.
- Given all this, I believe that the paper should not be accepted at ICLR.